# Description Generation Using Variational Auto-Encoders for Precursor microRNA

**DOI:** 10.3390/e26110921

**Published:** 2024-10-30

**Authors:** Marko Petković, Vlado Menkovski

**Affiliations:** 1Department of Applied Physics and Science Education, Eindhoven University of Technology, 5612AZ Eindhoven, The Netherlands; m.petkovic1@tue.nl; 2Eindhoven Artificial Intelligence Systems Institute, 5612AZ Eindhoven, The Netherlands; 3Department of Mathematics and Computer Science, Eindhoven University of Technology, 5612AZ Eindhoven, The Netherlands

**Keywords:** generative models, interpretability, description generation, microRNA

## Abstract

Micro RNAs (miRNA) are a type of non-coding RNA involved in gene regulation and can be associated with diseases such as cancer, cardiovascular, and neurological diseases. As such, identifying the entire genome of miRNA can be of great relevance. Since experimental methods for novel precursor miRNA (pre-miRNA) detection are complex and expensive, computational detection using Machine Learning (ML) could be useful. Existing ML methods are often complex black boxes that do not create an interpretable structural description of pre-miRNA. In this paper, we propose a novel framework that makes use of generative modeling through Variational Auto-Encoders to uncover the generative factors of pre-miRNA. After training the VAE, the pre-miRNA description is developed using a decision tree on the lower dimensional latent space. Applying the framework to miRNA classification, we obtain a high reconstruction and classification performance while also developing an accurate miRNA description.

## 1. Introduction

In living organisms, DNA encodes all the information used by the organism to develop and survive. To make use of the information encoded in the DNA, it needs to be translated to RNA. One type of RNA is messenger RNA (mRNA), which is used to create proteins. Sometimes, depending on the cell type, it is desirable that a certain mRNA is not expressed, which is where micro RNA (miRNA) comes into play by silencing and inhibiting the expression of certain mRNAs. MiRNAs can silence mRNAs by binding to them, following which the mRNA is degraded. As such, the definition of what constitutes an miRNA is functional since it needs to carry out this task. Depending on the shape of the miRNA, it can be used to silence (multiple) different mRNAs. Since over- and under-expression of miRNA can lead to various diseases, it is important to understand the genome of miRNA better.

It is estimated that there are around 2300 types of human miRNA [1] while only around 700 are documented [2]. Finding new miRNAs often requires complex lab conditions [3,4] where potential new miRNA needs to be differentiated from other RNA. To detect novel miRNAs, the focus is typically on identifying precursor miRNAs (pre-miRNAs), which are intermediate forms processed before becoming mature miRNAs. Initially, primary miRNA is transcribed from DNA and processed by enzymes into pre-miRNA [5]. Pre-miRNA is then further processed into mature miRNA within the cell. Folded in a hairpin structure, pre-miRNA contains more nucleotides and therefore possesses more distinguishing features (see Figure 1) compared to other types of RNA. As a result, detecting novel pre-miRNAs is generally easier than detecting mature miRNAs. While some features, like the presence of a terminal loop and a large fraction of base pairs (C-G/A-U) in the stem, are prevalent in pre-miRNA, they are not enough to differentiate it from other RNA molecules and hence are not sufficient to form a structural description of pre-miRNA.

A number of data-driven methods have been developed to identify novel pre-miRNA. Some of the pre-miRNA detection methods are based on traditional ML methods, which use engineered features [7] such as Random Forests (RFs) [8] and Support Vector Machines (SVMs) [3,8,9,10,11], where only limited interpretability is possible through feature importance. In addition, several Deep Learning methods have been proposed, which rely on Convolutional Neural Networks (CNNs) [12,13,14,15,16,17], Recurrent Neural Networks (RNNs) [14,18,19], and transformers [20,21]. While these methods are able to achieve state-of-the-art accuracy on the miRNA classification task, interpretation is only possible through methods that use activation maps or techniques such as concept whitening [22]. As such, these black-box models do not directly allow us to form a description of pre-miRNA, which would, in turn, allow us to better understand these molecules and their roles in the cell.

In recent years, generative models such as Variational Auto-Encoders (VAEs) and Generative Adversarial Networks (GANs) have been successfully applied to a wide variety of tasks such as protein [23,24], drug [25,26,27], and DNA design [28,29]. Typically, VAEs and GANs generate data by mapping a low-dimensional latent space to high-dimensional data. The low-dimensional latent space in these models is usually entangled, meaning that one latent variable can encode multiple aspects of the generative process. To tackle this problem, different disentanglement techniques have been proposed [30,31,32], where the loss function or model architecture forces the model to learn a latent representation in which each variable only encodes one aspect of the generative process. However, it is not inherently clear which latent variable corresponds to which property of the data and how these properties could be used to describe the class of a datapoint.

In this work, we develop a method for developing pre-miRNA descriptions, using the lower dimensional latent space of a latent variable model (Figure 2). Our contributions are two-fold: (i) We propose a novel framework based on VAEs and Decision Trees (DTs), which can be used to develop interpretable descriptions of the different object classes *y* being modeled in terms of features *f*; (ii) We apply this framework to the domain of precursor microRNA where we demonstrate that it obtains biologically sound structural descriptions.

## 2. Methods

### 2.1. Data

We use the *modmiRBase* dataset [12], which contains 49602 RNAs from various organisms and is balanced in terms of class. The pre-miRNA sequences in the dataset were sourced from mirbase.org [33] and mirgenedb.org [34]. The non-pre-miRNA sequences were obtained from various datasets, which included existing non-pre-miRNAs with characteristics similar to pre-miRNAs [3,35,36], as well as shuffled or suboptimally folded pre-miRNAs [7]. The RNA sequences are folded using the RNAFold algorithm [37], which predicts the secondary structure by identifying stable configurations where complementary nucleotides bond. Following this, the RNA sequences are encoded as images with dimensions of 100 by 25 pixels [12]. In this encoding, each nucleotide is represented as a colored bar—adenine (A) in blue, cytosine (C) in yellow, guanine (G) in green, and uracil (U) in red. Gaps (represented in black) are added for nucleotides lacking complementary pairs. The length of each bar is inversely related to the bond strength between nucleotides, with strong bonds occurring between base pairs and weaker bonds between other pairs or gaps. Additionally, the length of each bar increases with multiple consecutive weak bonds, while gaps are consistently represented with a length of 2 pixels. This encoding captures the physical structure of the RNA, with the hairpin formations resembling actual hairpins in the images. Additionally, any bulges present in the RNA structure are represented as protruding bars, accurately reflecting the spatial arrangement of the nucleotides. An example of an image encoding can be found in Figure 3.

Since the shape plays an important role in the process of silencing mRNAs, we also encode the bond strength between nucleotides. This is performed in vector *m* of size 100, where a strong bond is assigned a value of 1, while a weak bond (or no bond) is assigned a value of 0.

For each datapoint, we also calculated multiple features, using the algorithms from [22]. Several of the features can be found in Figure 1, such as the presence of a terminal loop, the size of the terminal loop, and the length of the stem. These features are not used directly in our VAE but are used in the decision tree algorithm instead.

### 2.2. Model

To model the (non) pre-miRNA, we made use of the Domain Invariant Variational Autoencoder (DIVA) framework [32], with a latent space for bond strength ($z_{m}$), class ($z_{y}$) and remaining variance ($z_{x}$). Each latent space has a size of 64. For each latent space, a separate encoder based on ResNet [38] was used, with an overview of the architecture in Figure 4a. Furthermore, $z_{m}$ and $z_{y}$ have conditional priors, whereas $z_{x}$ has a standard normal prior. The prior for $z_{m}$ consists of a 1D convolutional network (Figure 4b), followed by a fully connected layer, while the prior of $z_{y}$ consists of two fully connected layers. In addition, auxiliary classifiers were used to further enhance the disentanglement. The auxiliary classifier for m consists of a fully connected layer followed by 1D deconvolutional layers (Figure 4c), while the auxiliary classifier for *y* consists of two fully connected layers. Since the shape of the miRNA plays an important role in the mRNA silencing process, we made use of an additional auxiliary classifier for *y* on $z_{m}$. Finally, to organize the latent space better, Inverse Autoregressive Flow (IAF) [39] was applied to all 3 latent spaces. Each IAF consists of 8 blocks of 2 Masked Autoencoder for Distribution Estimation (MADE) [40] layers, which have a context size of 32 (obtained from the encoder), as well as a hidden size of 1080.

Since our data are highly structured, both the encoders and decoders were designed with the data complexity in mind. In the input representation, we one-hot encoded the 5 possible different colors to ensure that distances between different colors are always the same, which results in an image with 5 channels. Furthermore, using a regular deconvolutional image decoder could yield invalid reconstructions, as respecting the structure of the data would not be possible. To limit the amount of invalid images, we separately decode the height and color of each bar. After both height and color distributions are decoded, we combine the two distributions to obtain a distribution over both height and color, which is used to calculate the reconstruction loss. The decoder of the bar height is a 2D deconvolutional decoder. It outputs the probability for the different possible bar heights after softmax activation. To obtain the actual bars, the probability distribution is multiplied by the different bar heights, similar to the approach in StampNet [41]. The decoder for the bar color is a 1D deconvolutional decoder and outputs the color probabilities of the bars in the bottom and top rows. Finally, the heights and colors are multiplied to obtain the reconstruction. When sampling from the decoder, a discrete value should be sampled from both the height and color distributions. The architecture of both decoders can be found in Figure 4d. In both the encoders and decoders, the number of filters was chosen based on Cordero et al. [12].

An overview of the full model architecture can be found in Figure 5. In the model, each layer is followed by batch normalization and uses the ELU activation function (unless mentioned otherwise). For training the model, the loss functions in Equation (Equation 2) were used. In Equation (Equation 1), the first term represents the reconstruction loss, while the remaining terms represent the KL divergence between the encoded prior ($q_{\phi }$) and the (learned) prior ($p/p_{\theta }$). The penalty terms for each KL divergence are represented by β. In Equation (Equation 2), each additional term represents the loss of the different auxiliary classifiers ($q_{\omega }$), with α being the penalty term for each classifier. The dataset was split into a training set containing 34,721 samples and a test set containing 14,881 samples, with both datasets being balanced. For training, we set each β parameter to 0.5. Both $\alpha_{y1}$ and $\alpha_{y2}$ were set to 12, while $\alpha_{m}$ was set to 1. The hyperparameters for α and β were selected to ensure the model learns a well-organized latent space while still generating high-quality samples. The model was trained using the Adam optimizer [42] with a batch size of 64 and a learning rate of 0.0005 until convergence.
(1)$$\begin{array}{cc} L_{s}\left(m,x,y\right)=\; & E_{q_{\phi_{m}}\left(z_{m}|x\right),q_{\phi_{x}}\left(z_{x}|x\right),q_{\phi_{y}}\left(z_{y}|x\right)}\left[\log\left(p_{\theta }\left(x\;|\;z_{y},z_{x},z_{m}\right)\right)\right]\\ \; & -\beta_{m}*D_{KL}(q_{\phi_{m}}\left(z_{m}\;|\;x\right)\;||\;p_{\theta_{m}}\left(z_{m}\;|\;m\right))\\ \; & -\beta_{x}*D_{KL}(q_{\phi_{x}}\left(z_{x}\;|\;x\right)\;||\;p\left(z_{x}\right))\\ \; & -\beta_{y}*D_{KL}(q_{\phi_{y}}\left(z_{y}\;|\;x\right)\;||\;p_{\theta_{y}}\left(z_{y}\;|\;y\right)) \end{array}$$
(2)$$\begin{array}{cc} F_{\mathrm{DIVA}}\left(m,x,y\right) & =L_{s}\left(m,x,y\right)\\ \; & +\alpha_{y_{1}}E_{q_{\phi_{y}\left(z_{y}|x\right)}}\left[\log q_{\omega_{y}}\left(y\;|\;z_{y}\right)\right]\\ \; & +\alpha_{y_{2}}E_{q_{\phi_{y}\left(z_{m}|x\right)}}\left[\log q_{\omega_{y}}\left(y\;|\;z_{m}\right)\right]\\ \; & +\alpha_{m}E_{q_{\phi_{m}\left(z_{m}|x\right)}}\left[\log q_{\omega_{m}}\left(m\;|\;z_{m}\right)\right] \end{array}$$

To verify that the model architecture is optimal, we performed several ablation studies where we compared the aforementioned model with a standard(/β) VAE using a decoder consisting of fully connected layers (VAE), a β-VAE with IAF applied on the latent space (β-IAF-VAE), as well as with a deconvolutional decoder and IAF (DC-β-IAF-VAE). For all models, β was set to 0.5. Code for all models and experiments can be found on https://www.github.com/marko-petkovic/mirna (accessed on 22 September 2024).

### 2.3. Decision Tree

To develop a description of pre-miRNA, we create a decision tree algorithm on top of the $z_{m}$ latent space, as this latent space contains both information regarding the shape and class of RNA. Since this latent space is organized, various features of the RNA should be linearly separable within this space. As such, a linear SVM can be trained to partition the space based on a feature, such as the length of the RNA.

Compared to a normal decision tree algorithm, such as CART [43] or C4.5 [44], our algorithm makes splits based on the learned latent representation rather than the features (*f*) themselves. To create a split using our decision tree algorithm, for each feature, we train an SVM to partition the latent space ($z_{m}$), with the feature being the target variable. In case a feature is continuous, we make multiple binary features out of it, where the feature is divided according to different thresholds. Alternatively, one could train a regression model rather than a classifier and decide the threshold for a binary feature afterward. For each split, we assess whether the performance of the SVM is above a threshold, and only features classified with an accuracy above the threshold are considered. Then, to decide which split to make, we asses which split yields the highest information gain based on the class of the RNA. The SVM used to make the split is stored as an attribute of the decision tree node. After training a full decision tree, we can assess which descriptions of pre-miRNA the model developed by following the paths down the tree that lead to a pre-miRNA classification. The full algorithm for making splits for the DT can be found in Algorithm 1.
**Algorithm 1 MakeSplit**(z,y,f)**Require:** max_depth,min_samples,min_acc 1:Node←Node(z,y,concepts) 2:**if** current depth ≥max_depth **or** len(y)<min_samples **or** Node is pure **then**
 3:   **return** Node 4:**end if** 5:gain ← 0, cls ← None 6:**for** feature in *f* **do** 7:    train SVM (X=z,y=feature) 8:    split z based on classifier predictions in $z_{0}$, $z_{1}$ 9:    calculate information gain based on split 10:  **if** information gain > gain **and** SVM accuracy >min_acc **then** 11:      gain←informationgain 12:      cls ← SVM 13:  **end if** 14:**end for** 15:Split data into $z_{0}$, $z_{1}$, $f_{0}$, $f_{1}$, $y_{0}$, $y_{1}$ according to cls 16:Node.left_child ←**MakeSplit**($z_{0}$,$y_{0}$,$f_{0}$,depth) 17:Node.right_child ←**MakeSplit**($z_{1}$,$y_{1}$,$f_{1}$,depth) 18:**return** Node

To predict whether a new datapoint is a pre-miRNA, it should first be encoded by the encoder for $z_{m}$. Then, the decision tree should be followed, where it is checked on which side of the split by the SVM the point lies.

After training the model, we trained a decision tree with max_depth=5, min_samples=10, and min_acc=0.8 using the training set and features *f* from [22]. Each of the continuous features was partitioned into multiple binary features.

## 3. Results

### 3.1. Performance

In Table 1, the reconstruction performance of our model and the different ablations can be found. The reconstruction performance was calculated based on three different metrics: the Mean Absolute Error (MAE) of the entire image, the nucleotide accuracy, and the MAE of the length of the nucleotides. In Equations (Equation 3)–(Equation 5), the evaluation metrics are defined. In Equation (Equation 3), $y_{i}$ represents the true pixel value, while ${\hat{y}}_{i}$ is the predicted pixel value for each of the *P* pixels. In Equation (Equation 4), $c_{b}$ is the true color label, and ${\hat{c}}_{b}$ is the predicted nucleotide for the *b*-th bar, with *B* being the total number of bars. The indicator function $1(c_{b}={\hat{c}}_{b})$ evaluates to 1 if the colors match and 0 otherwise. Finally, in Equation (Equation 5), $L_{b}$ and ${\hat{L}}_{b}$ represent the true and predicted lengths of the *b*-th bar, respectively, for all *B* bars. As can be seen, the DC-IAF-DIVA model achieved the highest performance in terms of all three metrics.
(3)$$\begin{array}{cc} \mathrm{MAE} & =\frac{1}{P}\sum_{i=1}^{P}\left|y_{i}-{\hat{y}}_{i}\right| \end{array}$$
(4)$$\begin{array}{cc} \mathrm{Nucleotide}\;\mathrm{Accuracy} & =\frac{1}{B}\sum_{b=1}^{B}1\left(c_{b}={\hat{c}}_{b}\right) \end{array}$$
(5)$$\begin{array}{cc} \mathrm{MAE}\;\mathrm{length} & =\frac{1}{B}\sum_{b=1}^{B}\left|L_{b}-{\hat{L}}_{b}\right| \end{array}$$

To verify that the model has learned an organized latent space, we have visualized the latent spaces using t-SNE [45] dimensionality reduction. As can be seen in Figure 6, the disentanglement of the DIVA model indeed appears to be successful, as there seems to be a clear separation between classes in $z_{m}$ and $z_{y}$, while this is not present in $z_{x}$. Taking a further look at the distribution of the fraction of base pairs in the stem, terminal loop length, and the stem length over the $z_{m}$ latent space (Figure 7), we see that the latent space is well organized. Therefore, it can be used by our decision tree algorithm to develop a pre-miRNA description.

### 3.2. Conditional Generation

In addition to inspecting the latent space embeddings, we also qualitatively verify the learned representation of $z_{m}$. Here, we first encode a selected RNA image to obtain $z_{m}$, $z_{y}$, and $z_{x}$. Following this, the bond strength *m* of the sample is modified and passed through the conditional prior to obtain $z_{m}^{\prime }$. Afterward, we reconstruct the RNA based on $z_{m}^{\prime }$, $z_{y}$, and $z_{x}$. Finally, we verify whether the newly generated RNA image indeed follows the provided bond strength.

In Figure 8, we can find the results of the conditional generation experiment. In the top left, the original image can be found, with the bond strength *m* displayed below the picture. In each column, a different sample from the same conditional prior is reconstructed, while for each row a unique *m* was used. Overall, we see that there are a few (minor) errors in reconstruction, but in general, the generated images follow the provided bond strength *m*. As such, we can conclude that the learned latent space is well organized both in terms of features and bond strength.

### 3.3. Decision Tree

The obtained description from the decision tree can be found in Figure 9. Overall, the model achieved an accuracy of 0.912, a sensitivity of 0.923, and a specificity of 0.955. The concept whitening method achieved an accuracy of 0.928 on the same dataset while offering less interpretation compared to the decision tree. An overview of the performance comparison between the two methods can be found in Table 2.

In addition to the high classification performance, the decision tree learned biologically sound classification rules. The first split in the decision tree is made based on the amount of base pairs in the stem, where samples with a high amount of base pairs are considered to be pre-miRNA, which is also supported by literature [6]. Similarly, short RNA sequences are also considered unlikely to be pre-miRNA [6], which can be seen in the second split of the tree. The final feature used by the tree to discriminate pre-miRNA from other RNA is the absence of large asymmetric bulges, which is also a known property of pre-miRNA [46]. In general, given the supplied features, we see that the decision tree is able to generate a biologically valid description of pre-miRNA.

## 4. Discussion

In this work, we have proposed a new method for latent space interpretability in VAEs and applied it to pre-miRNA generation. The trained VAE for pre-miRNA generation has shown excellent performance, validating its suitability for the application of our interpretability method. Our analysis demonstrated that the latent space descriptions developed through the method align with existing biological knowledge of pre-miRNAs, suggesting both the VAE model and our interpretability approach perform robustly in capturing meaningful structural features.

Despite these promising results, there are limitations to our approach. The model is focused solely on pre-miRNA structures and does not account for interactions with target genes, which could provide further insights into miRNA functionality. Additionally, the interpretability method relies on existing biological knowledge for validation, meaning its utility may be limited in less well-characterized regions of the latent space or for novel pre-miRNA variants. Expanding the model to account for more diverse and less-studied pre-miRNA types could pose challenges in interpretability and generalization.

A potential direction for future work could be to explore the use of geometric GNNs instead of CNN-based models for learning pre-miRNA structures. Since miRNA is primarily distinguished from other RNA by its function, which is closely linked to its shape, GNNs may provide a more effective representation of the spatial and structural properties of miRNAs. By using a geometric representation of the RNA molecule, GNNs can more accurately capture and process its shape, potentially improving interpretability and generalization across more complex datasets compared to image-based representations. Additionally, target interaction data, such as miRNA-mRNA interactions, could be integrated to account for functional relationships that influence miRNA activity. Recent machine learning methods that predict miRNA-target interactions [47,48,49] could be leveraged to enhance the model’s understanding of these regulatory networks. Incorporating such interaction data, either as additional input features or for refining the latent space, could significantly improve the model’s ability to generalize across diverse datasets and capture the biological relevance of pre-miRNAs more effectively.

## Figures and Tables

**Figure 1 entropy-26-00921-f001:**
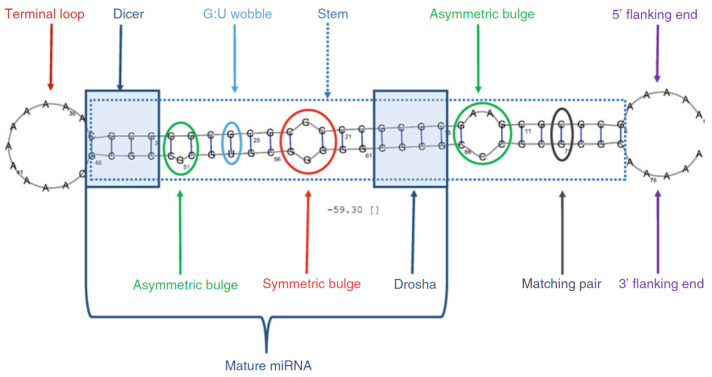
Artificial pre-miRNA strand [6] labeled with different properties that can be present in (non) pre-miRNA.

**Figure 2 entropy-26-00921-f002:**
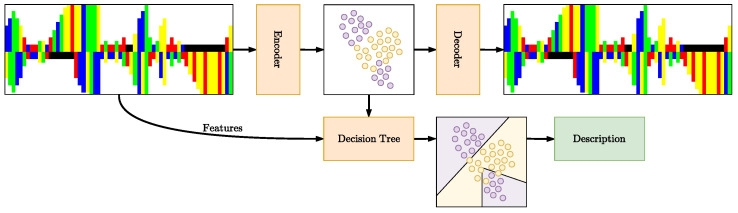
Overview of the proposed description generation framework. The Variational Autoencoder (VAE) is trained to encode the input into a latent space, which is then decoded. A Decision Tree (DT) is trained on the latent space to split based on features, using an SVM for each feature to assess which split maximizes information gain for the classification of pre-miRNA versus non-pre-miRNA. The resulting tree provides interpretable descriptions of pre-miRNA.

**Figure 3 entropy-26-00921-f003:**
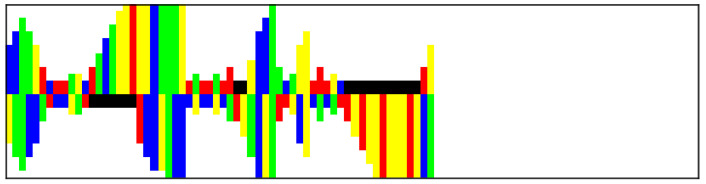
Example of RNA image encoding.

**Figure 4 entropy-26-00921-f004:**
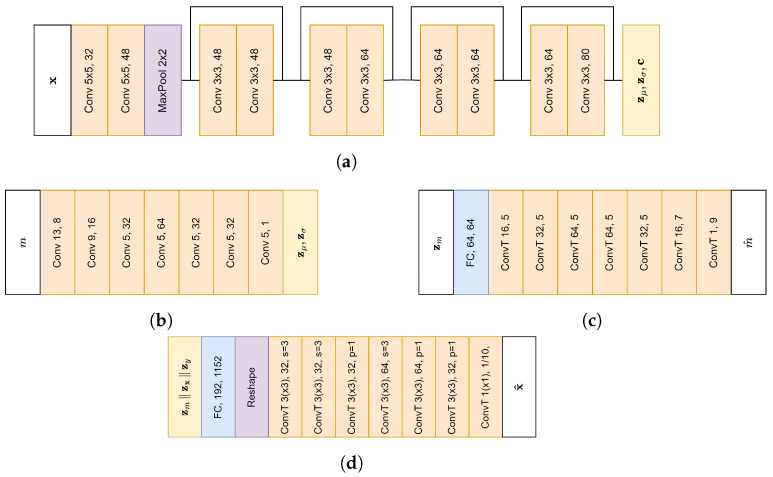
Architecture of different model components. The first number following the layer name indicates the kernel size, and the second number indicates the amount of filters. Unless mentioned otherwise, each convolutional layer uses a stride (s) of 1. Convolutional layers in the encoder use same padding (p), while other layers use valid padding unless mentioned otherwise. (**a**) Encoder architecture. Sampling from the final layer is followed by IAF to obtain the latent space. (**b**) Prior ($z_{m}$) architecture. (**c**) Classifier for *m* architecture. (**d**) Decoder for bar height/color. Both decoders are preceded by the same fully connected layer and reshape operation. The height decoder uses 2D transposed convolutions, while the decoder for color first sums over the height dimension and then uses 1D transposed convolutions. For the final layer, the color decoder gives the probability for each color (top and bottom row, 5 colors in each bar). Outputs of the two decoders are combined to obtain $\hat{x}$.

**Figure 5 entropy-26-00921-f005:**
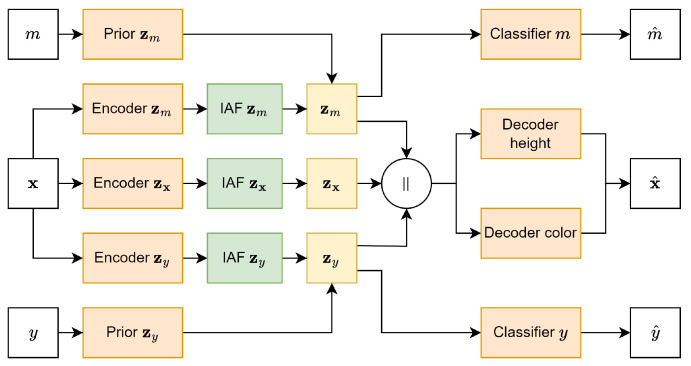
Full DIVA model architecture. ∥ represents the concatenation of the different latent spaces.

**Figure 6 entropy-26-00921-f006:**
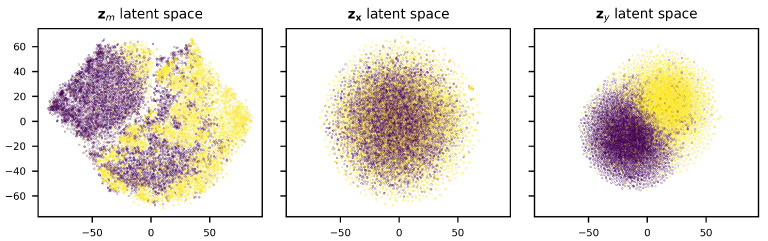
Latent spaces of DIVA model after dimensionality reduction using t-SNE. Yellow dots represent pre-miRNA.

**Figure 7 entropy-26-00921-f007:**
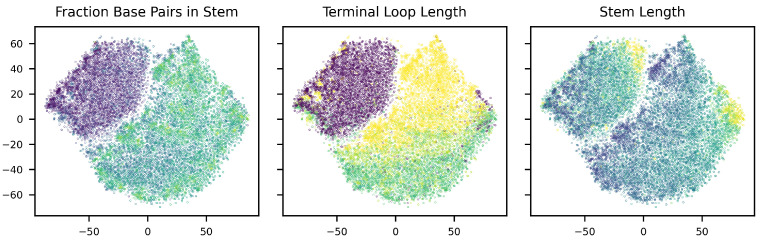
$z_{m}$ latent space colored according to the fraction base pairs in stem, terminal loop length, and stem length. Lighter colors indicate higher values for each variable.

**Figure 8 entropy-26-00921-f008:**
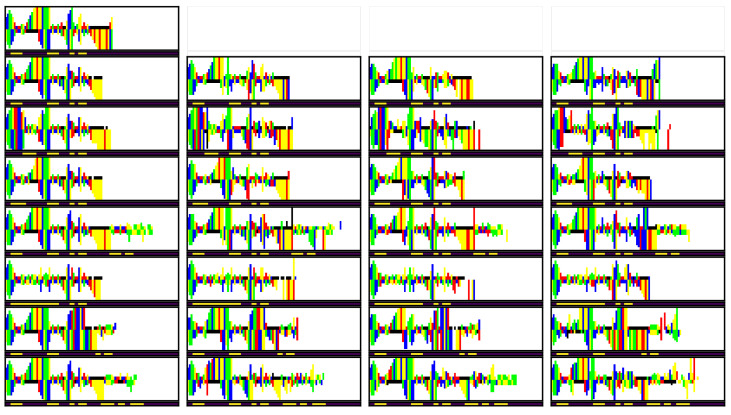
Generated RNA images, with $z_{m}$ sampled from the prior of various *m*. In each row, a different *m* was used. In the second row, the same *m* as in the original image was used.

**Figure 9 entropy-26-00921-f009:**
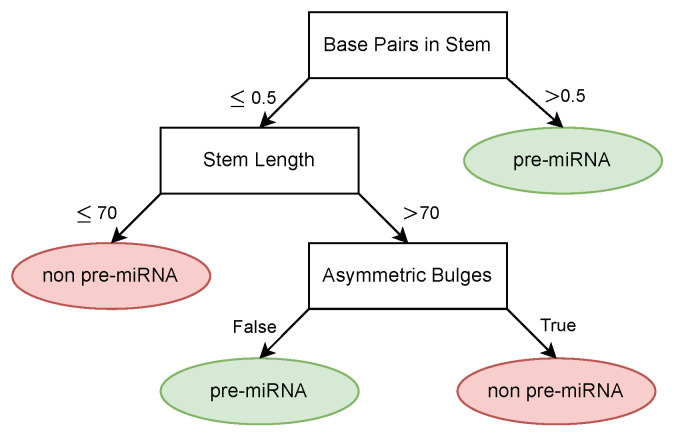
Learned pre-miRNA description.

**Table 1 entropy-26-00921-t001:** Reconstruction statistics for different model types. Bold statistics represent the lowest obtained errors.

Model Name	MAE	Nucleotide Accuracy	MAE Length
VAE	0.136	0.695	0.784
β-VAE	0.055	0.908	0.478
β-IAF-VAE	0.053	0.918	0.444
DC-β-IAF-VAE	0.009	0.985	0.067
DC-IAF-DIVA	**0.007**	**0.988**	**0.057**

**Table 2 entropy-26-00921-t002:** Model Performance on pre-miRNA Classification.

Model	Accuracy	Sensitivity	Specificity
Decision Tree VAE	0.912	0.923	0.955
Concept Whitening [22]	0.928	-	-

## Data Availability

The data as well as the code for the experiments and results can be found on https://github.com/marko-petkovic/mirna (accessed on 22 September 2024).

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
