# Peer review of "Description Generation Using Variational Auto-Encoders for Precursor microRNA"

_entropy, 2024, doi:10.3390/e26110921_

Round 1
Reviewer 1 Report
Comments and Suggestions for Authors
In the work described in the manuscript, the authors introduce a novel computational framework for detecting and interpreting precursor microRNAs (pre-miRNAs).To overcome the complexity and lack of interpretability in existing machine learning methods, the authors employ Variational Auto-Encoders (VAEs) to uncover the generative factors of pre-miRNAs. They then use a decision tree on the latent space to develop an interpretable description. The framework achieves high reconstruction and classification performance, providing an accurate and interpretable method for miRNA detection.
However, this work has some weaknesses, in my opinion, that require a revision.
1. There are some grammatical errors in the manuscript. The author is advised to carefully review and make necessary corrections.
2. Some of the figures in the manuscript are of low resolution, which affects their clarity and readability. I recommend replacing these images with higher-resolution versions to ensure that all details are clearly visible and to enhance the overall quality of the presentation.
3. Include more experimental details, such as dataset sources, and preprocessing methods to help readers better understand the experimental design and results.
4. The manuscript mentions choices for different hyperparameters, such as beta values and learning rates, but lacks detailed justification for these choices. It is recommended that the authors add an explanation of how to tune these hyperparameters to enhance the reproducibility and transparency of this work.
5. In a future work section, provide a more detailed plan of how you intend to enhance the generalization capabilities of the model and integrate additional biological data to build a comprehensive model. In addition, it might be better to discuss more explicitly the advantages that GNNs may exhibit in representing RNA structures over current methods.
6. In order to enhance the credibility and scientific basis of the paper, I propose to cite some key bioinformatics literature in the paper, especially those related to miRNA detection, RNA interactions, or interpretable machine learning in bioinformatics,, such as doi:10.1016/j.asoc.2024.111523, doi:10.1007/s11432-024-4098-3, and 10.1186/s12859-023-05309-w, and so on.
Comments on the Quality of English LanguageMinor editing of English language required.
Author Response
Reviewer
In the work described in the manuscript, the authors introduce a novel computational framework for detecting and interpreting precursor microRNAs (pre-miRNAs).To overcome the complexity and lack of interpretability in existing machine learning methods, the authors employ Variational Auto-Encoders (VAEs) to uncover the generative factors of pre-miRNAs. They then use a decision tree on the latent space to develop an interpretable description. The framework achieves high reconstruction and classification performance, providing an accurate and interpretable method for miRNA detection.
Reply
Thank you for the valuable feedback. We have carefully reviewed the manuscript to correct grammatical errors and replaced Figure 1 with a higher-resolution version. Additionally, we have expanded the experimental details and provided more explanation for our choice of hyperparameters. In response to the suggestion on future directions, we have included a discussion on using geometric GNNs for RNA structure representation and integrating additional biological data. Lastly, we have added the recommended citations to strengthen the scientific foundation of our work. We hope these revisions address your concerns.
Reviewer
- There are some grammatical errors in the manuscript. The author is advised to carefully review and make necessary corrections.
Reply
We have reviewed the manuscript, and corrected grammatical errors. We appreciate the reviewer for pointing this out.
Reviewer
- Some of the figures in the manuscript are of low resolution, which affects their clarity and readability. I recommend replacing these images with higher-resolution versions to ensure that all details are clearly visible and to enhance the overall quality of the presentation.
Reply
Thank you for pointing this out. We have replaced Figure 1 with a higher resolution versions.
Reviewer
- Include more experimental details, such as dataset sources, and preprocessing methods to help readers better understand the experimental design and results.
Reply
Revisions have been made to clarify the experimental design, including additional details about the dataset sources and preprocessing methods. These updates should provide readers with a clearer understanding of the experimental setup and the results obtained. The dataset section of the methods has been updated on lines 70 – 86.
Reviewer
- The manuscript mentions choices for different hyperparameters, such as beta values and learning rates, but lacks detailed justification for these choices. It is recommended that the authors add an explanation of how to tune these hyperparameters to enhance the reproducibility and transparency of this work.
Reply
We appreciate the reviewer's suggestion to clarify the choices for the hyperparameters, and we have made changes in the manuscript to provide additional detail. The convolutional filters used in the encoder and decoder were adopted from prior work, as they have proven effective for similar tasks. The values for hyperparameters such as α\alphaα and β\betaβ were selected to balance the trade-off between achieving an organized latent space and generating high-quality samples. This ensures that both aspects are captured adequately. For the learning rate, we empirically determined that the chosen value provided stable training and yielded good results for this specific task.
While a more thorough exploration of hyperparameter tuning was not the primary focus of this work, we found our choices to be sufficient for achieving our objectives. Future work could explore systematic tuning methods to further improve reproducibility. These changes have been reflected in the revised text for transparency on lines 128-129 and 140-141.
Reviewer
- In a future work section, provide a more detailed plan of how you intend to enhance the generalization capabilities of the model and integrate additional biological data to build a comprehensive model. In addition, it might be better to discuss more explicitly the advantages that GNNs may exhibit in representing RNA structures over current methods.
Reply
We agree with the reviewer that more explicit discussion on how to enhance the model's generalization capabilities and the advantages of GNNs would be valuable. In response, we have added a discussion in the revised manuscript about the potential of using geometric GNNs instead of CNN-based models for learning pre-miRNA structures. Since miRNA function is closely tied to its shape, GNNs could offer a more effective representation of the spatial and structural properties, leading to improved interpretability and generalization compared to image-based methods.
Furthermore, we suggest incorporating additional biological data such as RNA secondary structure (e.g., base pair probabilities and stem-loop regions from RNAFold) and expression patterns from public databases to create a more comprehensive model. These features could enhance the model’s ability to generalize across diverse datasets. We hope this aligns with the reviewer's feedback and provides a clearer vision for future work. We have updated the future work section to include this information (lines 243 – 255).
Reviewer
- In order to enhance the credibility and scientific basis of the paper, I propose to cite some key bioinformatics literature in the paper, especially those related to miRNA detection, RNA interactions, or interpretable machine learning in bioinformatics,, such as doi:10.1016/j.asoc.2024.111523, doi:10.1007/s11432-024-4098-3, and 10.1186/s12859-023-05309-w, and so on.
Reply
We have added additional citations regarding generative AI in bioinformatics in lines 51-52. Additionally, we have cited works on miRNA target detection in line 251.
Reviewer 2 Report
Comments and Suggestions for Authors
The authors presented an interesting method to detect and classify by using Variational Auto-Encoders through the projection of the predicted pre-microRNA structures into colour coded images. microRNA detection and characterisation has significance in many biomedical applications and it is welcome to see efforts in tackling this difficult task. The paper in general is well written and easy to follow. I have a couple of comments.
Firstly, the mechanism of colour coding the RNA structures is not explained in this paper. I understand some of this information may be found in one of the literature, but given this idea serves as one of the fundamentals of the method, it is better to describe at least briefly in the paper.
Secondly, the idea is illustrated in the method section. It would be informative to provide a little more details on the second part of the algorithm regarding the decision tree classification, so that the paper becomes more approachable to general readers.
Thirdly, in result section, Table 1. gives important statistics of the performance. Authors need to describe more clearly the methods compared in the main texts or in the figure caption. It would be also interesting to see the result comparison of the proposal method to other more traditional ML models.
Author Response
Reviewer
The authors presented an interesting method to detect and classify by using Variational Auto-Encoders through the projection of the predicted pre-microRNA structures into colour coded images. microRNA detection and characterisation has significance in many biomedical applications and it is welcome to see efforts in tackling this difficult task. The paper in general is well written and easy to follow. I have a couple of comments.
Reply
Thank you for your insightful comments. We have clarified the mechanism of color coding RNA structures, explaining how nucleotides (A, U, C, G) are assigned specific colors to allow the model to process the RNA as an image. Additionally, we have expanded the method section to provide more details on the decision tree classification process, including how SVMs are used for splits in the latent space. Regarding Table 1, we have added clearer descriptions of the compared methods and updated the figure caption. While we acknowledge the interest in comparing with traditional ML models, the focus of our work is on generating descriptive representations, and we believe this contribution stands on its own.
Reviewer
Firstly, the mechanism of colour coding the RNA structures is not explained in this paper. I understand some of this information may be found in one of the literature, but given this idea serves as one of the fundamentals of the method, it is better to describe at least briefly in the paper.
Reply
We agree that explaining the color coding process is important, since it serves as one of the fundamentals of the model. Through the color coding, we assign a color to each type of nucleotide (A,U,C,G), such that the model can process the image representation of the RNA. We have further clarified this mechanism in the introduction in lines 70-86.
Reviewer
Secondly, the idea is illustrated in the method section. It would be informative to provide a little more details on the second part of the algorithm regarding the decision tree classification, so that the paper becomes more approachable to general readers.
Reply
We appreciate the reviewer's suggestion to provide more detail on the decision tree classification in the method section. In response, we have expanded the explanation to clarify how splits are made in the latent space using SVMs for each feature (lines 167). We also provided additional information on how continuous features are handled and partitioned into binary features based on thresholds (lines 164-165). We hope this makes the algorithm more accessible for general readers and enhances the overall clarity of the method.
Reviewer
Thirdly, in result section, Table 1. gives important statistics of the performance. Authors need to describe more clearly the methods compared in the main texts or in the figure caption. It would be also interesting to see the result comparison of the proposal method to other more traditional ML models.
Reply
We have clarified the descriptions of the methods in the main text and figure caption to enhance their comprehensibility. While we appreciate the suggestion to compare our proposed method with traditional machine learning models, it's important to note that achieving high performance is not the primary focus of this paper. Our main objective is to develop descriptive representations, and we believe this contribution stands independently of direct comparisons with conventional ML models.
Reviewer 3 Report
Comments and Suggestions for Authors
This paper proposes a VAE-DT-based framework for generating interpretable descriptions of precursor microRNA (pre-miRNA) using a combination of variational autoencoders and decision trees. The proposed method introduces two key innovations: the utilization of a disentangled VAE latent space to uncover relevant structural features of pre-miRNA and the application of a decision tree to generate biologically meaningful descriptions. This framework integrates both image-based representations of RNA sequences and bond strength features to enhance miRNA classification and interpretability, incorporating a latent variable model to separate key factors such as shape and base pairing. While the model demonstrates significant improvements in pre-miRNA classification tasks, achieving 91.2% accuracy, there are still several issues that need to be addressed.
1. The paper claims to address a binary classification problem for pre-miRNA prediction, yet the keyword section does not include any mention of pre-miRNA, which is critical for the study's focus.
2. In the Introduction, the structure requires refinement. The first paragraph contains excessive information on miRNA, while providing insufficient details about pre-miRNA, the paper's central topic. Moreover, there is a noticeable lack of coverage regarding previous work, and many of the cited references are outdated, mostly from before 2020. A more comprehensive review of recent literature is needed.
3. The paper contains various minor issues that need attention. For instance, some abbreviations are introduced without clarification—DIVA in Section 2.2 (Model) is not defined in full. Additionally, figures are not placed in the most relevant sections. For example, Figure 8 is referenced in Section 3.2, yet it only appears in Section 3.3, which disrupts the flow of the presentation.
4. Figure 3, which represents the workflow of the proposed model, needs improvement as it does not intuitively reflect the contributions and methodology of this paper. It is suggested to include symbol legends and provide explanations for all the annotations in the figure to improve clarity.
5. The paper lacks a comprehensive ablation study. While it compares various VAE variants, it does not examine the overall impact of using VAE itself. Similarly, the contribution of the decision tree classifier has not been compared with other algorithms, and the roles of DIVA and IAF in the model's performance are also unexplored.
6. The paper contains only one table, which presents the ablation study of various VAE variants, but it lacks additional tables to provide a more comprehensive comparison. Furthermore, the key prediction results are not organized into a table format for clarity and ease of reference; instead, they are described in text within Section 3.3.
7. The paper lacks a detailed description of the evaluation metrics used in the study. It would be beneficial to introduce the metrics clearly and explain how they are calculated to assess the model's performance.
Comments on the Quality of English Language
No question
Author Response
Reviewer
This paper proposes a VAE-DT-based framework for generating interpretable descriptions of precursor microRNA (pre-miRNA) using a combination of variational autoencoders and decision trees. The proposed method introduces two key innovations: the utilization of a disentangled VAE latent space to uncover relevant structural features of pre-miRNA and the application of a decision tree to generate biologically meaningful descriptions. This framework integrates both image-based representations of RNA sequences and bond strength features to enhance miRNA classification and interpretability, incorporating a latent variable model to separate key factors such as shape and base pairing. While the model demonstrates significant improvements in pre-miRNA classification tasks, achieving 91.2% accuracy, there are still several issues that need to be addressed.
Reply
Thank you for the constructive feedback. In response to your suggestions, we have added pre-miRNA to the keyword section and refined the introduction to focus more on pre-miRNA and its role in miRNA detection. Additionally, we have expanded the literature review to include more recent references. We have fully defined all abbreviations and reorganized the figures to improve the flow of the paper. Furthermore, we have replaced Figure 3 with a more intuitive workflow representation and clarified its annotations. While we acknowledge the importance of a comprehensive ablation study, the focus of this paper is on interpretability rather than optimizing model performance. However, we have added a table summarizing key prediction results and clarified the evaluation metrics, providing formulas for metrics like MAE and nucleotide accuracy to improve transparency. We hope these changes address your concerns and enhance the clarity and rigor of the manuscript.
Reviewer
- The paper claims to address a binary classification problem for pre-miRNA prediction, yet the keyword section does not include any mention of pre-miRNA, which is critical for the study's focus.
Thank you for pointing this out. We have added pre-miRNA as a keyword.
Reviewer
- In the Introduction, the structure requires refinement. The first paragraph contains excessive information on miRNA, while providing insufficient details about pre-miRNA, the paper's central topic. Moreover, there is a noticeable lack of coverage regarding previous work, and many of the cited references are outdated, mostly from before 2020. A more comprehensive review of recent literature is needed.
Reply
We agree that the first paragraph mainly focuses on miRNA, whereas insufficient details about pre-miRNA are provided. Pre-miRNA is processed by enzymes into mature miRNA, by cutting of certain parts of the nucleotide sequence. Since pre-miRNA has more distinctive features, it is easier to differentiate from other RNA, and is therefore used in the computational detection of miRNA. We have included additional information in the introduction to more clearly highlight why we are using pre-miRNA, and how it relates to miRNA on lines 28-34.
We have expanded the literature review to include more recent methods on Deep Learning for (pre-)miRNA detection. This was updated in the introduction on lines 44 – 45.
Reviewer
- The paper contains various minor issues that need attention. For instance, some abbreviations are introduced without clarification—DIVA in Section 2.2 (Model) is not defined in full. Additionally, figures are not placed in the most relevant sections. For example, Figure 8 is referenced in Section 3.2, yet it only appears in Section 3.3, which disrupts the flow of the presentation.
Reply
Thank you for the suggestion. We have fully defined all the abbreviations, and have reordered the figures to ensure a smooth flow of the presentation.
Reviewer
- Figure 3, which represents the workflow of the proposed model, needs improvement as it does not intuitively reflect the contributions and methodology of this paper. It is suggested to include symbol legends and provide explanations for all the annotations in the figure to improve clarity.
Reply
We agree that (old) Figure 3 does not adequately represent the contributions of this work, as it merely shows the architecture of the DIVA model. We have inserted a new figure (Figure 3), which more intuitively reflects our proposed methodology. In addition, we have updated the caption of (old) Figure 3 to improve the clarity.
Reviewer
- The paper lacks a comprehensive ablation study. While it compares various VAE variants, it does not examine the overall impact of using VAE itself. Similarly, the contribution of the decision tree classifier has not been compared with other algorithms, and the roles of DIVA and IAF in the model's performance are also unexplored.
Reply
Our method generates descriptions by utilizing the lower-dimensional, organized latent space of a generative model, which necessitates the use of VAEs. Similarly, the use of decision trees (DT) is essential for generating interpretable rules for miRNAs, as other methods typically offer feature importance but do not directly provide rule-based descriptions. While we have not fully investigated the specific contributions of DIVA and IAF, our primary focus was on creating an organized latent space, and these models were sufficiently effective for this purpose. The emphasis of our work was not on achieving the best possible model performance but rather on interpretability and structure.
Reviewer
- The paper contains only one table, which presents the ablation study of various VAE variants, but it lacks additional tables to provide a more comprehensive comparison. Furthermore, the key prediction results are not organized into a table format for clarity and ease of reference; instead, they are described in text within Section 3.3.
Reply
A table has been added to present the key prediction results, focusing on prediction performance for clarity and ease of reference. Table 1 primarily highlights how well the VAE variants balance the creation of an organized latent space while maintaining a strong generative process. Although we did not include additional comparisons with other model types, future work could explore a more comprehensive analysis across various metrics and models to provide further insights.
Reviewer
- The paper lacks a detailed description of the evaluation metrics used in the study. It would be beneficial to introduce the metrics clearly and explain how they are calculated to assess the model's performance.
Reply
We thank the reviewer for their comment. In response, we have added a more detailed explanation of the evaluation metrics used to assess the model's performance, including the specific formulas for calculating the mean absolute error (MAE), nucleotide accuracy, and length MAE. The MAE measures the reconstruction error by comparing the true and predicted pixel values across all images. Nucleotide accuracy evaluates how accurately the model predicts the nucleotide types, using an indicator function to check if the predicted nucleotide matches the true nucleotide. Length MAE calculates the error between the true and predicted lengths of the bars in the generated images. The detailed formulas (Equations 3-5) and explanations of the symbols have been included in the manuscript (lines 181-188) to enhance the clarity and reproducibility of the evaluation process.
Round 2
Reviewer 1 Report
Comments and Suggestions for Authors
The author responded to my question, but there are still issues that need to be resolved to meet the publishing requirements. The author only uploaded the code of the model on GitHub without creating a web service that is convenient for readers to use, which is not enough. Although the author cited some bioinformatics articles, it is still necessary to increase the citation of the latest relevant articles to improve the readability of the article. The author analyzed and discussed some of the results obtained by the proposed model in computational experiments. Can case studies be added to verify the usability of the model?
Comments on the Quality of English LanguageMinor editing of English language required.
Reviewer 3 Report
Comments and Suggestions for Authors
No questions